# Highly Sensitive Sub-ppm CH_3_COOH Detection by Improved Assembly of Sn_3_O_4_-RGO Nanocomposite

**DOI:** 10.3390/molecules27248707

**Published:** 2022-12-08

**Authors:** Norazreen Abd Aziz, Mohd Faizol Abdullah, Siti Aishah Mohamad Badaruddin, Mohd Rofei Mat Hussin, Abdul Manaf Hashim

**Affiliations:** 1Faculty of Engineering & Built Environment, Universiti Kebangsaan Malaysia, Bangi 43600, Malaysia; 2MIMOS Semiconductor (M) Sdn Bhd, Technology Park Malaysia, Kuala Lumpur 57000, Malaysia; 3Malaysia-Japan International Institute of Technology, Universiti Teknologi Malaysia, Jalan Sultan Yahya Petra, Kuala Lumpur 54100, Malaysia

**Keywords:** Sn_3_O_4_ nanoparticles, reduced graphene oxide, heterojunction, CH_3_COOH

## Abstract

Detection of sub-ppm acetic acid (CH_3_COOH) is in demand for environmental gas monitoring. In this article, we propose a CH_3_COOH gas sensor based on Sn_3_O_4_ and reduced graphene oxide (RGO), where the assembly of Sn_3_O_4_-RGO nanocomposites is dependent on the synthesis method. Three nanocomposites prepared by three different synthesis methods are investigated. The optimum assembly is by hydrothermal reactions of Sn^4+^ salts and pre-reduced RGO (designated as RS nanocomposite). Raman spectra verified the fingerprint of RGO in the synthesized RS nanocomposite. The Sn_3_O_4_ planes of (111), (210), (130), ($\bar{13}2$) are observed from the X-ray diffractogram, and its average crystallite size is 3.94 nm. X-ray photoelectron spectroscopy on Sn3d and O1s spectra confirm the stoichiometry of Sn_3_O_4_ with Sn:O ratio = 0.76. Sn_3_O_4_-RGO-RS exhibits the highest response of 74% and 4% at 2 and 0.3 ppm, respectively. The sensitivity within sub-ppm CH_3_COOH is 64%/ppm. Its superior sensing performance is owing to the embedded and uniformly wrapped Sn_3_O_4_ nanoparticles on RGO sheets. This allows a massive relative change in electron concentration at the Sn_3_O_4_-RGO heterojunction during the on/off exposure of CH_3_COOH. Additionally, the operation is performed at room temperature, possesses good repeatability, and consumes only ~4 µW, and is a step closer to the development of a commercial CH_3_COOH sensor.

## 1. Introduction

Various sensors for detecting and monitoring toxic and harmful gases including volatile organic compounds (VOCs) have been studied to satisfy the regulation of the environmental and gases industry standards. Several significant methods have been used so far to monitor VOCs and other toxic gases including electrochemical sensors [1], acoustic sensors [2], optical sensors [3], and colorimetric sensors [4]. Acetic acid (CH_3_COOH), which is one of the most corrosive and highly irritating VOCs, is broadly used in food, plastic, pharmaceutical, and most manufacturing industries [5,6]. There can be many potential health issues including eye and skin irritation, body swelling, particularly the nose, tongue, throat, etc., and respiratory illnesses from prolonged exposure to high concentrations of CH_3_COOH. According to guidelines from China and the USA, the maximum allowable concentrations of acetic acid vapor in the workplace are 7.5 ppm (GBZ 2-2002) and 9.3 ppm (D3620-04), respectively [7]. Therefore, developing an ultrasensitive, low limit of detection, low energy consumption, and reliable acetic acid gas sensor is highly desirable for air monitoring in modern industries and workplaces.

Compared to conventional CH_3_COOH detection analysis such as gas chromatography and mass spectrometry, chemiresistive metal oxide gas sensors are more suitable for real-time analysis because of their advantages including compact size, easy production, and simple measuring electronics [8,9,10,11]. SnO_2_, ZnO, and TiO are the most common metal oxide materials which have been extensively used for detecting various kinds of gases due to their high sensitivities, low costs, simple fabrications, and long-term stabilities. In addition to SnO_2_ [12,13,14,15], other tin oxides with other oxygen stoichiometries such as SnO [16,17,18], Sn_2_O_3_ [19], Sn_3_O_4_ [20,21,22], and Sn_5_O_6_ [23], have been widely investigated. Among them, Sn_3_O_4_, which has a band gap of ~2.9 eV, has attracted significant attention owing to its unique sensing properties. However, these single-material-based gas sensors typically need to be operated at high temperatures to obtain a desirable sensing response, which leads to considerably high power consumption, a scarce lifetime, and poor selectivity of the device [24,25]. Undeniably, there is also a recent work that reported on the successful operation at room temperature of a triethylamine (TEA) gas sensor based on porous SnO_2_ films with rich oxygen vacancies [26].

To overcome the above-mentioned drawbacks, the heterostructure that is constructed by incorporating metal oxide semiconductor (MOS) materials with innovative graphene-based materials [27,28,29,30] can be a valid alternative. Graphene and its derivatives, especially reduced graphene oxide (RGO) possess several promising features, including a large specific surface area for better adsorption, thermoelectric conductivity, and high carrier mobility at room temperature [31,32]. Generally, RGO under ambient conditions exhibits p-type behavior due to its electron-withdrawing nature, and if the incorporated metal oxide behaves as an n-type semiconductor, the formation of p-n heterostructures may be realized. The p-n heterostructures can modulate the electronic and chemical properties via surface charge transfer and chemical bonding, thus improving the gas sensing performance.

Numerous studies have been conducted to improve the gas sensing characteristics through the formation of metal oxide-RGO nanocomposites including CuO-RGO [33], SnO_2_-RGO [34,35], ZnO-RGO [36], TiO_2_-RGO [37], and many more. Although the accurate analysis of all the effective parameters is still not clear, the major factors that influence the sensing performance are the assembly of metal oxide nanoparticles with RGO, deoxygenation, and the number of active sites, i.e., oxygen vacancies, O_v_ and chemisorbed oxygen, O_c_ available for capturing targeted analytes [38]. From previous work, different microstructures are observed from the family of tin oxide-RGO composites that are prepared by various synthesis techniques, thus leading to different sensing performances [39,40]. To date, only a few studies on metal oxide-RGO-based CH_3_COOH sensors have been reported [41,42]. Therefore, we proposed a highly sensitive sub-ppm CH_3_COOH sensor based on Sn_3_O_4_-RGO nanocomposite operating at room temperature.

In this work, we investigate the effect of hydrothermal synthesis conditions on the material microstructures and their sensing performances to CH_3_COOH. An exhaustive study on three types of Sn_3_O_4_-RGO sensing materials that were prepared via (i) hydrothermal reactions of Sn^4+^ salts and pre-reduced RGO (designated as RS nanocomposite) (ii) one-step hydrothermal synthesis from Sn^4+^ and GO (designated as OS nanocomposite), (iii) Sn_3_O_4_ nanoparticles cast onto RGO nanosheet (designated as CO nanocomposite) to validate the effects of synthesis condition on microstructure and other sensing properties.

## 2. Materials and Methods

### 2.1. Materials

The reagents used to synthesize the Sn_3_O_4_-RGO nanocomposites include a commercial (4 mg/mL) graphene oxide, GO (dispersion in H_2_O), and SnCl_4_·5H_2_O (99%) purchased from Sigma Aldrich. Both were used as received, without any further purification.

#### 2.1.1. Preparation of One-Step Sn_3_O_4_-RGO Nanocomposite (OS Nanocomposite)

Figure 1a shows the typical one-pot Sn_3_O_4_-RGO nanocomposite that was synthesized via the facile hydrothermal method. A total of 187.5 µL of (GO) from the bottle and 24 mg of SnCl_4_·5H_2_O were dissolved in 20 mL of deionized water under magnetic stirring for 1 h to achieve homogenous dispersion. Subsequently, the dispersion was transferred into a 50 mL Teflon-lined stainless-steel autoclave and was heated in the oven at a temperature of 180 °C for 12 h. After cooling down to room temperature, the solution was centrifuged at 4000 rpm for 15 min. At last, the OS nanocomposite was obtained by washing three times with deionized water and one time with ethanol.

#### 2.1.2. Preparation of Cast-On Sn_3_O_4_-RGO Nanocomposite (CO Nanocomposite)

CO nanocomposite was obtained by assembling Sn_3_O_4_ nanoparticles into GO nanosheets through hydrothermal treatment as shown in Figure 1b. Firstly, 24 mg of SnCl_4_·5H_2_O was dissolved into 20 mL of deionized water with constant magnetic stirring for 1 h. The solution was poured into a 50 mL Teflon-lined stainless-steel autoclave and then heated in the oven at a temperature of 180 °C for 12 h. The resulting Sn_3_O_4_ nanoparticles were obtained by centrifuge washing four times using deionized water and ethanol. The resulting Sn_3_O_4_ NPs were poured into 20 mL of deionized water together with 187.5 µL GO and then stirred again for another 1 h. At last, the mixture underwent the same hydrothermal process (180 °C for 12 h) and was washed four times with water and ethanol subsequently to achieve the CO nanocomposite.

#### 2.1.3. Preparation of Pre-Reduced Sn_3_O_4_-RGO Nanocomposite (RS Nanocomposite)

To achieve RS nanocomposite, a two-step hydrothermal treatment is needed, starting with the thermal reduction of GO as shown in Figure 1c. At first, 187.5 µL of GO was dispersed into 20 mL of deionized water under vigorous stirring for 1 h. Then the mixture was poured into a 50 mL Teflon-lined stainless-steel autoclave for hydrothermal reduction treatment (180 °C for 12 h). The pre-reduced RGO dispersion in deionized water was obtained. Then 24 mg of SnCl_4_·5H_2_O was added to the RGO solution and continued by the same process, i.e., stirring for 1 h and heating at 180 °C for 12 h. Finally, the RS nanocomposite was collected by washing it four times with water and ethanol.

### 2.2. Material Characterization

The crystal structure and phase composition of each as-prepared sample were evaluated by the X-ray diffraction (XRD) system (PANalytical Empyrean diffractometer with Cu K*α* radiation: *λ* = 1.5418 Å). The oscillation modes of each composite were investigated using 473 nm laser Raman spectroscopy (NTEGRA Spectra MT-MDT). The microstructure and morphology of Sn_3_O_4_-RGO nanocomposites were inspected by transmission electron microscope (TEM) with an accelerating voltage of 120 kV (Talos, L120C from Thermo Fisher Scientific, Eindhoven, The Netherlands). The composition of S, O, and C were analyzed by X-ray photoemission spectroscopy (XPS) system (NEXSA G2 from Thermo Fisher Scientific) using a monochromated Al K*α* source (voltage source: 12 kV, beam size: 100 µm, pass energy survey scan: 200 eV, pass energy narrow scan: 100 eV).

### 2.3. Sensor Fabrication and Measurement Set-Up

The gas sensor was established by dropping 5 µL aqueous dispersion of different sensing materials prepared previously on a 1 × 1 cm^2^ Si/SiO_2_ substrate where interdigitated electrodes (IDE) were fabricated on it. The Pt/Ti (100/10 nm) IDE geometry consists of 3 pairs of finger electrodes and each electrode has a width and length of 0.1 mm and 2 mm, respectively. The interfinger spacing between electrodes is 0.2 mm. This test structure was constructed using lithography, metallization, and a standard lift-off process. Thus, the coverage of the sensing area was ~6 mm^2^. The prepared gas sensing devices were dried overnight in the desiccator. Figure 2 shows a schematic of the measurement setup that was used to investigate the sensing performance of the three different composites towards CH_3_COOH. In an enclosed chamber, the sensor was placed on the heater stage and controlled by Nextron Temperature Controller. The temperature was set at room temperature. The flow rates of compressed dry air (CDA) and CH_3_COOH were controlled by mass flow controllers of the humidifier system (Cellkraft P-10). The relative humidity was set constant at 5 ± 1% using a closed-loop control system. The testing was performed by applying a direct voltage of 2 V and monitoring changes in the electrical resistance (using source measuring unit Keithley 2410). The sensor response, *R_s_* is quantified by changes in resistance upon exposure to CH_3_COOH analytes and air (CDA in this case) as shown in Equation (1).
(1)$$R_{s}=[(R_{g}-R_{a})/R_{a}]\times 100\%$$
where *R_g_* is the resistance in the presence of CH_3_COOH gas and *R_a_* is the gas sensor’s electrical resistance at rest when exposed to CDA. In the meantime, the response time is described as the time needed for the CH_3_COOH sensor to respond at 90% of its saturated level and the recovery time is the amount of time required to recover 10% from its initial value when exposed to the air. The sensitivity (unit = %/ppm) of the Sn_3_O_4_-RGO sensor is calculated from the change of *R_s_* for the varied concentration of acetic acid. To be fair, we set the durations of the on/off flow of acetic acid as constant for all measurements. To record the response and recovery speed, we started to record the measurement as soon as CDA was introduced in the chamber. After 90 s, CDA was automatically shut off and CH_3_COOH was released in the chamber immediately.

## 3. Results and Discussion

### 3.1. Sn_3_O_4_-RGO Nanocomposites Formation Mechanism

We chose the hydrothermal method to synthesize Sn_3_O_4_-RGO-based composites as their advantages inhomogeneous distribution and versatile interfacial decorations between RGO and metal oxide. Three kinds of Sn_3_O_4_-RGO-based nanocomposites were made by changing the preparation condition via a hydrothermal method using GO and SnCl_4_.5H_2_O as precursors. During the synthesis process, there are several reactions that were carried out under the hydrothermal temperature and pressure in the autoclave i.e., (i) the adsorption of Sn^4+^ by GO or RGO, (ii) the reduction of GO into RGO, and (iii) Sn^4+^ decomposing into Sn_3_O_4_ nanocrystals. The formation chemistry of the Sn_3_O_4_-RGO nanocomposites is explained as follows:(2)$${\mathrm{SnCl}}_{4}\cdot 5H_{2}O\to {\mathrm{Sn}}^{4+}+{\mathrm{Cl}}^{-}+5H^{+}+5{\mathrm{OH}}^{-}$$
(3)$${\mathrm{Sn}}^{4+}+{4\mathrm{OH}}^{-}\to {\mathrm{Sn}(\mathrm{OH})}_{4}$$
(4)$${3[\mathrm{Sn}(\mathrm{OH})}_{4}]\to {\mathrm{Sn}}_{3}O_{4}+6H_{2}O+O_{2}$$

Although maintaining the volume of GO and SnCl_4_·5H_2_O, the amount of Sn_3_O_4_ nanostructures, RGO obtained, and also the nanocomposite interfaces may be different due to the tuning of synthesis methods.

### 3.2. Structural and Morphological Analysis

Figure 3 depicts the XRD patterns ranging from 20° to 60° of all three nanocomposites. Notably, all as-prepared nanocomposites show three strong diffraction peaks at 2θ of 26.41°, 33.82°, and 51.73° which are attributed to (111), (210), (130), and ($\bar{1}\bar{3}2$) planes of triclinic Sn_3_O_4_ structure (JCPDS Card No 16-0737) [43], indicating the successful formation of the single crystal Sn_3_O_4_ particles in this work. The full width at half maximum (FWHM) of all peaks is broad, revealing a nanoscale size of Sn_3_O_4_. The crystallite sizes can be determined by the Debye–Sherrer formula as follows; *D* = 0.89*λ*/*β*cos *θ*, where *D* is the average crystallite size diameter, *λ* (Cu K*α*) is 0.154 Å, and *β* is the FWHM of the diffraction curves. The calculated average crystallite size values are 3.91, 3.38, and 3.94 nm for OS, CO, and RS nanocomposites, respectively. The relatively small Sn_3_O_4_ crystallite size of the CO nanocomposites is because Sn_3_O_4_ particles underwent two times the hydrothermal treatment as compared to the other method. Thus, the observed XRD results confirm the formation of nanosized Sn_3_O_4_-RGO nanocomposites. There is no peak observed for the RGO, which implies that the formation of the RGO plane maybe not be in perfect stacking and needs to be confirmed by Raman analysis.

The three as-prepared samples were then further characterized by the Raman technique as it is a useful method to characterize the structure of RGO-based materials. Figure 4 shows the Raman spectra of all three OS, CO, and RS nanocomposites. It can be deduced that all samples exhibit two major peaks at ~1300 cm^−1^ and ~1600 cm^−1^, corresponding to the characteristic D-band (*I_D_*) and G-band (*I_G_*), respectively. The *I_D_* is related to structural defects and partially disordered structure, and the *I_G_* refers to the vibration of sp^2^-bonded carbon atoms. The intensity ratio of the D and G bands (*I_D_/I_G_*) not only gives information about in-plane crystallite size, *L_a_*, but also indicates the overall degree of disorder within the graphitic carbon [44]. The average value of the *I_D_/I_G_* ratio and *L_a_* for all three nanocomposites are 1.07 and 2.93 nm singly. This condition indicates a high disorder of carbon sp^2^ atoms leading to high deoxygenation that concludes that GO is successfully reduced by hydrothermal preparation. It is also understood that the rich defects of OS nanocomposite are possibly due to an increase in vacancies, grain boundaries, and amorphous carbon species, as well as the insertion of Sn_3_O_4_ nanoparticles into RGO sheets [45]. This Raman scattering analysis demonstrates the existence of RGO in all three Sn_3_O_4_-RGO nanocomposites.

The morphology and surface nature of the as-prepared nanocomposites was characterized by TEM. Figure 5 depicts low-, medium-, and high-magnification TEM images and histograms of the three nanocomposites. From low-magnification TEM images, it can be seen that a huge amount of spherical Sn_3_O_4_ nanoparticles were formed on massive wrinkled RGO sheets. Most Sn_3_O_4_ nanoparticles were uniformly dispersed onto RGO sheets but only a small amount of them aggregate at the edge of OS nanocomposite (Figure 5a). Likewise, in the RS nanocomposite, Sn_3_O_4_ nanoparticles are evenly and densely distributed, only some Sn_3_O_4_ nanoparticles agglomerate at different spots on RGO sheets (Figure 5j). Meanwhile, a large amount of Sn_3_O_4_ nanoparticles aggregated on RGO sheets and we also observed the absence of Sn_3_O_4_ nanoparticles on the edge of CO nanocomposites (Figure 5e). From medium magnification, it is confirmed that there is good dispersity of Sn_3_O_4_ nanoparticles on the RGO sheet for OS and RS nanocomposites (Figure 5b,j). Meanwhile, for CO nanocomposites, it is confirmed there is an island of aggregated Sn_3_O_4_ nanoparticles isolated on the edge of some spare place on RGO as shown in Figure 5f. These conditions suggest that OS and RS are the best preparation method for Sn_3_O_4_-RGO nanocomposites as Sn_3_O_4_ nanoparticles prefer to grow all over the surface of GO and RGO by pre-adsorbing Sn^4+^ during the hydrothermal synthesis process. High-magnification TEM images as shown in Figure 5c,g,k indicate a clear image of Sn_3_O_4_ nanoparticles with non-uniform size distribution loading on the surface of RGO sheets. From histogram distribution (Figure 5d,h,l) the average particle sizes of Sn_3_O_4_ of OS, CO, and RS nanocomposites are 1.81, 1.77, and 1.75 nm, respectively. Thus, they are indicating that single crystal Sn_3_O_4_ is forming clusters of polycrystalline Sn_3_O_4_ nanoparticles.

XPS characterization was examined to investigate elements, surface chemistry, and electronic structure of Sn_3_O_4_-RGO nanocomposites. The survey spectra in Figure 6a indicate the peaks of Sn4d, Sn3d, Sn3p, O1s, and C1s that are observed in the region of 0–1200 eV. Thus, it confirms the presence of three elements of Sn, O, and C in all three nanocomposites. The peaks of Sn and O are much higher than the peak of C, implying the formation of a large amount of Sn_3_O_4_ nanoparticles. The components of all these elements are tabulated in Table 1. It is found that the ratio of tin and oxygen in these three nanocomposites is around 0.75, revealing the formation of Sn_3_O_4_ nanoparticles. The ratio of Sn_3_O_4_:RGO for samples OS, CO, and RS are 2.07, 4.62, and 3.60, respectively. These match with the TEM images in Figure 5, where high values of Sn_3_O_4_:RGO from samples CO and RS are due to the dispersion of Sn_3_O_4_ nanoparticles agglomerated onto the RGO sheet. Figure 6b displays the Sn3d spectrum of all three nanocomposites. It is obviously seen that two strong peaks at 487.3 eV and 495.7 eV attribute to the spin–orbit coupling of Sn3d5/2 and Sn3d3/2, respectively.

Meanwhile, Figure 6c shows the deconvoluted peaks of the C1s spectrum which exhibits three types of the carbon-associated group including C-C (~284.6 eV), C-O (~285.6 eV), and C=O (~289.2 eV). The components of these band groups are presented in Table 2. Notably, the relative percentage of the C-C band is much higher than that of the C-O and C=O group in both RS and CO nanocomposites, indicating the successful reduction of GO by the hydrothermal process. It can be concluded the reduction degree of RGO in RS nanocomposite is the strongest because the RGO underwent hydrothermal treatment two times. Figure 6d represents the wide and asymmetric feature of O1s which can be divided into four peaks, revealing the presence of four types of O-related species including the Sn-O-Sn lattice O atoms in triclinic Sn_3_O_4_ (O_L_), oxygen vacancies (O_v_), chemisorbed oxygen-related species such as hydroxyl groups (O_c_), and the C=O band at the binding energies of ~530.8, ~531.2, ~531.7, and ~532.6 eV, respectively. As can be seen from summarized data in Table 3, the atomic percentages of O_c_ and O_v_ are considerably high for both RS (O_c_ = 17.0%, O_v_ = 16.85%) and CO (O_c_ = 17.41%, O_v_ = 18.06%) nanocomposite, which is deemed to enhance the gas sensing response to CH_3_COOH. All these observations reveal the successful formation of all three Sn_3_O_4_-RGO nanocomposites synthesized hydrothermally by three different conditions.

### 3.3. Sensing Response

The sensing properties of the as-prepared nanocomposites sensor were tested for different sub-ppm concentrations of CH_3_COOH gas. The gas concentration varied from 0.3 to 6 ppm and the testing was carried out under room temperature at normal atmospheric pressure. Before the gas testing, the current–voltage characteristics of all Sn_3_O_4_-RGO-based sensors were measured by SMU Keithley 2410. As can be seen in Figure 7a, all three devices showed linear I–V characteristics over the voltage range from 0 to 10 V, which indicated Sn_3_O_4_-RGO is typical of ohmic electrical contact with electrodes. The corresponding initial resistance and operating power values of devices with CO, OS, and RS sensing materials are 1.24 MΩ, 19.23 kΩ, and 79.5 MΩ, and 3.24 µW, 0.2 mW, and 0.81 µW, respectively. During the sensing, the average power consumption increased to an average of 3.6 µW, 0.25 mW, and 3.9 µW, respectively. Plots for resistance for each device can be found in the Appendix A. The high conductivity of the CO device is due to enormous Sn_3_O_4_ nanoparticles which were also confirmed by XPS analysis (Table 1) that could possibly behave similar to metallic wires on RGO sheets.

Figure 7b,c illustrates the response and recovery curves over time for each CH_3_COOH concentration. The response and recovery time are defined by the time taken by a sensor to achieve 90% of the total resistance change. From the response curve, it can be clearly observed that generally, all three devices showed faster response with increasing concentration, although at a low concentration of below 1 ppm all three sensors provide a slow response rate. This could be attributed to the lower coverage of CH_3_COOH gas molecules to react with adsorbed oxygen ions, hence the change in resistance also takes place slowly. Comparatively, all three sensors could recover to their original condition with a fast recovery rate at a low concentration below 1 ppm. A good sensor should have a high value of response and a fast value for response and recovery time. The Sn_3_O_4_-RGO-RS sensor presents a more remarkable sensing performance as it exhibits a fast response and shorter recovery time compared to the other two sensors.

Figure 8a–f demonstrates the individual transient response and recovery curves of the three sensors exposed to various CH_3_COOH concentrations. Each exposure and recovery cycle is carried out in a 90 s interval under gas exposure followed by a recovery interval of 90 s under dry air conditions. Each cycle is indicated by the white area in the graph marking the start and end. Clearly, the corresponding response values of the lowest concentration 0.3 ppm are about 1.4% (*t_res_* = 66 s, *t_rec_* = 41 s) and 0.4% (*t_res_* = 46 s, *t_rec_* = 32 s) detected by OS nanocomposite and CO nanocomposite, respectively. The fluctuating response in Sn_3_O_4_-RGO-RS (4%) might be due to local carrier annihilations from the recombination of the electron with hole puddles in RGO. On the other hand, it can be clearly observed that the response of the RS nanocomposite sensor is much higher than that of the OS and CO sensors. When the sensor is exposed to the concentrations of 0.5, 1, 2, 4, and 6 ppm, the corresponding response values are about 25.9%, 52%, 72.3%, 73%, and 73.4% respectively.

The response curve of Sn_3_O_4_–RGO-based gas sensors to various sub-ppm CH_3_COOH concentrations was plotted in Figure 9a. Obviously, the response of all three sensors revealed an exponential trend at the beginning and reached saturation at a higher sub-ppm concentration, making 2 ppm the upper limit of detection. Variation of the response as a function of concentration is termed sensitivity, *S*, which is one of the most important criteria in gas sensing. The RS nanocomposite showed the highest sensitivity of 65.4%/ppm among the other two, OS nanocomposites (*S* = 3.03%/ppm) and CO nanocomposite (*S* = 0.45%/ppm). Figure 9b depicts the dynamic transient sensing response of Sn_3_O_4_-RGO-RS to 1 ppm CH_3_COOH. It is unambiguous that the Sn_3_O_4_-RGO-RS sensor can maintain consistent response/recovery characteristics during the four cycles of the test, demonstrating good repeatability of the device.

A comparison of previously reported gas sensors based on sensing materials in terms of their sensing properties toward CH_3_COOH is summarized in Table 4. In contrast to other materials, our Sn_3_O_4_-RGO-based sensor shows excellent CH_3_COOH sensing performances, including very high response and low limits of detection with acceptable response and recovery rate. Moreover, it provides an effective and simple method for the development of a CH_3_COOH gas sensor with good sensitivity and repeatability at room temperature.

### 3.4. Sensing Mechanism

The above results demonstrated the high-performance CH_3_COOH sensing properties of the Sn_3_O_4_-RGO sensor. As we know, Sn_3_O_4_ is an n-type semiconductor [41] and RGO nanosheets synthesized using chemical or low thermal treatments exhibit p-type semiconductor characteristics [52]. We proposed two possible sensing mechanisms: (i) the adsorption–desorption pathway on the surface of Sn_3_O_4_ nanoparticles, and (ii) the formation of Sn_3_O_4_-RGO heterojunction. Figure 10a illustrates a schematic of the boundary barrier model of Sn_3_O_4_ grains. When the CH_3_COOH sensor is exposed to clean air, the oxygen molecules in the atmosphere are adsorbed onto the surface of the Sn_3_O_4_ and then capture free conduction electrons creating oxygen anions (O^−^, O_2_^−^) on the surface of the material. This process forms a depletion layer (DL), reduces carrier concentration at the junction, and therefore, displays a high resistance reading. When the sensor is exposed to the targeted gas, the CH_3_COOH molecules desorb, or remove the oxygen anions, from the material’s surface. As a result, tremendous free electrons are released back into the conduction band of Sn_3_O_4_, thereby narrowing the electron depletion layer 1 (DL1) and reducing the resistance value.

On another note, based on the TEM image (Figure 5e), most of the Sn_3_O_4_ nanoparticles are not uniformly distributed on the RGO, and prefer to clump and overlap each other, which is also confirmed by the XPS data where there is an abundant O_v_ and O_c_ observed in the CO nanocomposite. However, due to their tendency to clump together, the absorption surface becomes smaller than it should be. This could be the reason for poor contact between RGO and Sn_3_O_4_ nanoparticles, leading to its poor sensing performances compared to the other two nanocomposites. The possible interaction involved in CH_3_COOH gas sensing is as follows [45]:(5)$$O_{2}\;(\mathrm{gas})+2e^{-}\to 2O^{-}\;(\mathrm{ads})$$
(6)$${\mathrm{CH}}_{3}\mathrm{COOH}\;(\mathrm{ads})+4O^{-}\;(\mathrm{ads})\to 2{\mathrm{CO}}_{2}+2H_{2}O+4e^{-}$$

Figure 10b reveals another underlying mechanism for the Sn_3_O_4_-RGO sensor as it introduces a heterojunction between Sn_3_O_4_ and RGO. Generally, when the sensor is exposed to the air, three types of electron DL could possibly be formed. DL1 formed on the surface of Sn_3_O_4_ nanoparticles owing to the adsorbed oxygen ions. Meanwhile, the formation of DL2 is by the electrons and transfers from the surface of Sn_3_O_4_ to the RGO during the formation of heterojunctions. By the third electron depletion, DL3 appears in the area where Sn_3_O_4_ is embedded in RGO nanosheets and forms the heterojunctions. All these electron depletion layers prevent the migration of electrons, resulting in a high-resistance state of the nanocomposites. When CH_3_COOH gas is introduced, the width of all these three DLs will undergo different changes. The trapped electrons in the DL1 will be released back to the Sn_3_O_4_, decreasing the width of the DL1. DL2 and DL3 are supposed to be narrowed because the reducing CH_3_COOH gas molecules may be adsorbed on the surface of RGO and donate the electrons to the RGO lattice, leading to a reduced efficacy of the junction barrier. The Sn_3_O_4_-RGO-RS nanocomposite exhibited superior sensing properties mainly attributed to its spherical 3D nanostructure and its formation on the RGO sheet. Compared with the other two CO and OS nanocomposites, this sensing film has more embedded and uniformly wrapped Sn_3_O_4_ nanoparticles on RGO sheets (DL2 and DL3 condition), which is also confirmed by TEM images (Figure 5k) As a result, the magnitude of thermionic emission current is increased during CH_3_COOH exposure due to massive relative change in electron concentration.

## 4. Conclusions

In summary, three Sn_3_O_4_-RGO nanocomposites were successfully synthesized via three different facile assembly methods. A comparative analysis of nanocomposite assembly was carried out to investigate their performance on CH_3_COOH sensing properties. The Sn_3_O_4_-RGO-RS nanocomposite showed a higher response and sensitivity, faster response, and faster recovery behavior to CH_3_COOH compared with another two nanocomposites. The experimental data confirm a clear correlation between surface structures of Sn_3_O_4_ and distribution of Sn_3_O_4_ on RGO nanosheets, which influenced their sensing behavior towards CH_3_COOH. The present study provides an effective approach that could speed up the development of highly sensitive room temperature CH_3_COOH sensors.

## Figures and Tables

**Figure 1 molecules-27-08707-f001:**
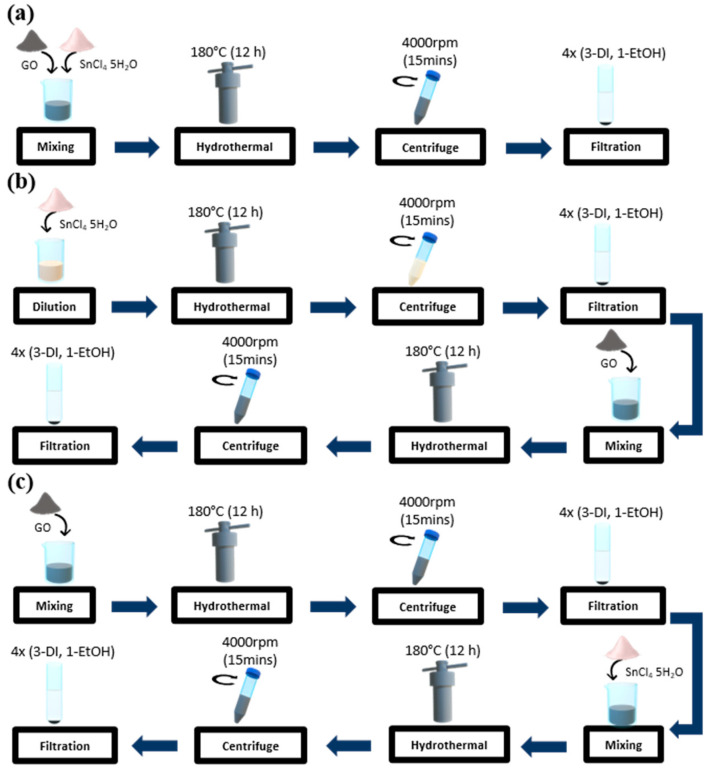
Schematic illustration of the preparation of three Sn_3_O_4_-RGO nanocomposites, (**a**) OS nanocomposite (**b**) CO nanocomposite, (**c**) RS nanocomposite with different synthesis conditions.

**Figure 2 molecules-27-08707-f002:**
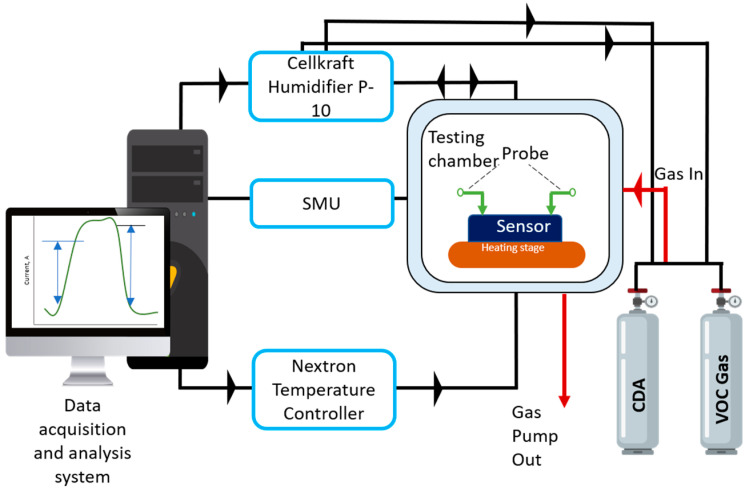
Schematic of illustration CH_3_COOH testing setup.

**Figure 3 molecules-27-08707-f003:**
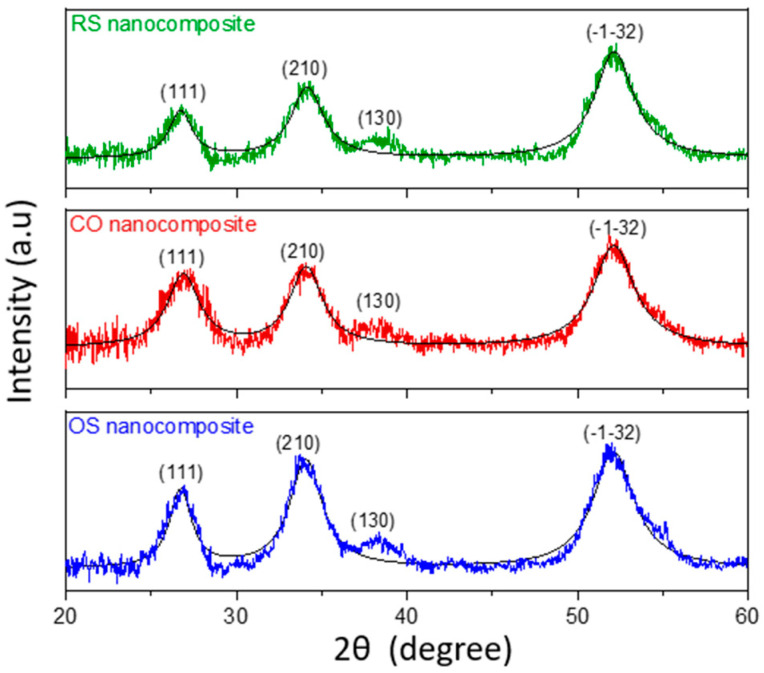
XRD diffractogram of OS, CO, and RS nanocomposites.

**Figure 4 molecules-27-08707-f004:**
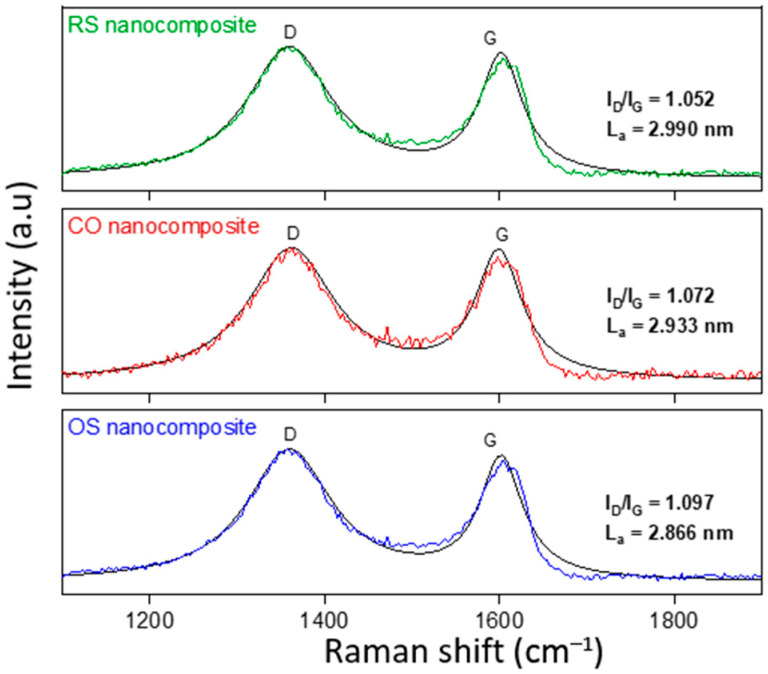
Raman spectra of OS, CO, and RS nanocomposite.

**Figure 5 molecules-27-08707-f005:**
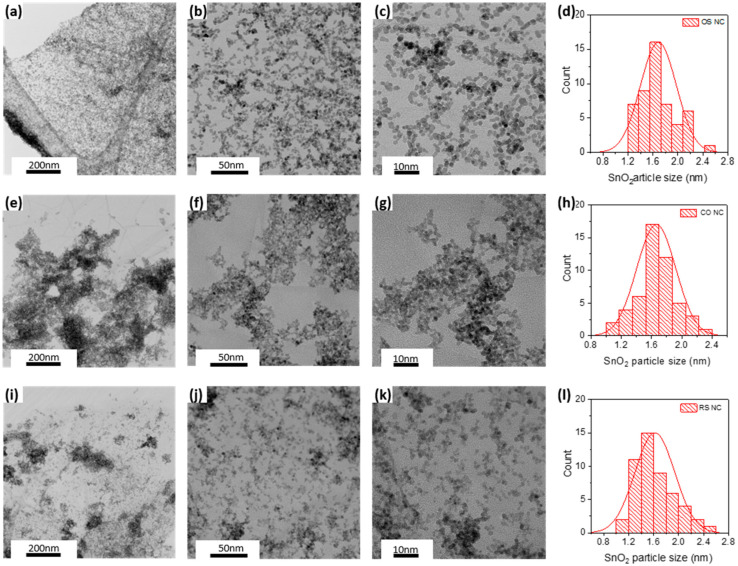
(**a**,**e**,**i**) Low magnification, (**b**,**f**,**j**) medium magnification, (**c**,**g**,**k**) high magnification images, and (**d**,**h**,**l**) Particle count histogram of Sn_3_O_4_ of OS, CO, and RS nanocomposites.

**Figure 6 molecules-27-08707-f006:**
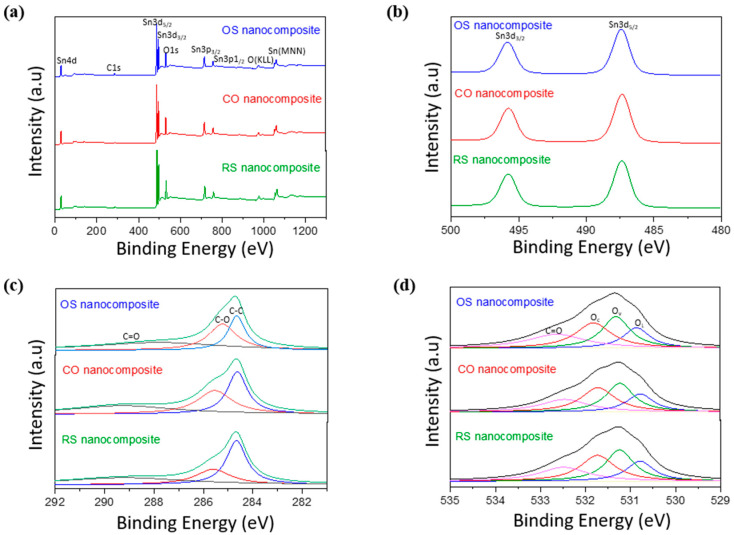
The (**a**) XPS spectra, (**b**) Sn3d XPS spectra, (**c**) C1s XPS spectra, and (**d**) O1s XPS spectra of OS, CO, and RS nanocomposites.

**Figure 7 molecules-27-08707-f007:**
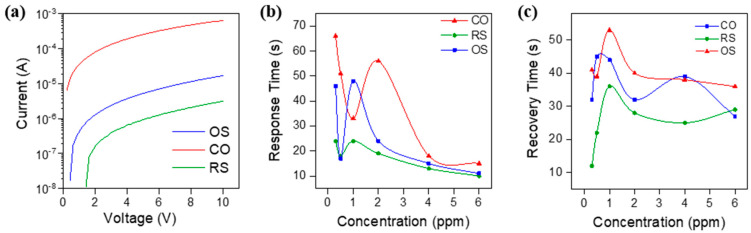
(**a**) I–V characteristic curve, (**b**,**c**) the response and recovery time of the Sn_3_O_4_-RGO-based sensor to various concentrations CH_3_COOH at room temperature.

**Figure 8 molecules-27-08707-f008:**
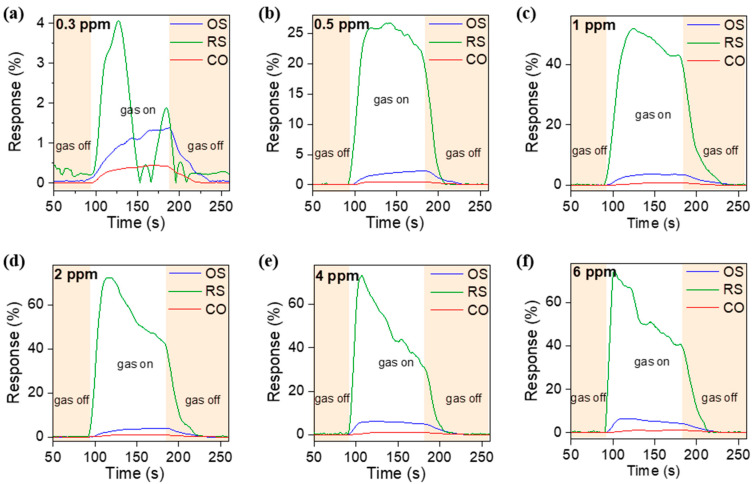
Real-time response–recovery curves of Sn_3_O_4_-RGO gas sensors under different CH_3_COOH concentrations: (**a**) 0.3 ppm, (**b**) 0.5 ppm, (**c**) 1 ppm, (**d**) 2 ppm, (**e**) 4 ppm, and (**f**) 6 ppm.

**Figure 9 molecules-27-08707-f009:**
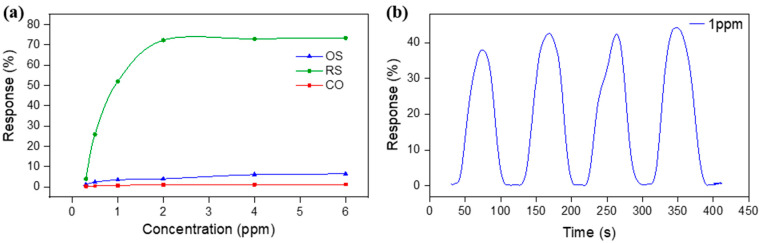
(**a**) Relationship between the sensing response of the Sn_3_O_4_-RGO-based sensor and CH_3_COOH concentration. (**b**) Dynamic transient response curves of the Sn_3_O_4_-RGO-RS sensor to 1 ppm CH_3_COOH during the four cycles of the test.

**Figure 10 molecules-27-08707-f010:**
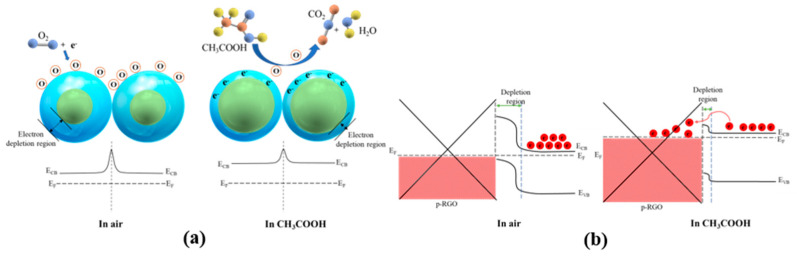
Schematic illustration of sensing mechanism of (**a**) Sn_3_O_4_-Sn_3_O_4_ nanoparticles, (**b**) Sn_3_O_4_-RGO nanocomposites.

**Table 1 molecules-27-08707-t001:** The fitting results of the content and content ratio of each atom for OS, CO, and RS nanocomposites.

Samples	Sn (At.%)	C (At.%)	O (At.%)	O* (At.%)	O** (At.%)	Sn:O** [Sn_3_O_4_]	C:O* [RGO]	Ratio Sn_3_O_4_:RGO
OS nanocomposite	29.15	19.58	51.27	12.95	38.32	0.76	1.51	2.07
CO nanocomposite	36.1	4.72	59.19	13.07	46.12	0.78	0.36	4.62
RS nanocomposite	33.89	8.88	57.23	12.85	44.38	0.76	0.70	3.60

O* = O atom in RGO (At.%). O** = O in Sn_3_O_4_ (At.%).

**Table 2 molecules-27-08707-t002:** The fitting results of C1s XPS spectrum of OS, CO, and RS nanocomposites.

Samples	Carbon Bonding	Binding Energy (eV)	At. (%)	Relative Percentage (%)
OS nanocomposite	C-C	284.6	5.35	27.33
	C-O	285.2	6.67	34.05
	C=O	287.8	7.55	38.60
CO nanocomposite	C-C	284.6	1.68	34.93
	C-O	285.5	1.57	33.25
	C=O	289.2	1.50	31.81
RS nanocomposite	C-C	284.6	3.69	41.61
	C-O	285.6	2.18	24.62
	C=O	289.1	2.99	33.76

**Table 3 molecules-27-08707-t003:** The fitting results of O1s XPS spectrum of OS, CO, and RS nanocomposites.

Samples	Oxygen Species	Binding Energy (eV)	At (%)	Relative Percentage (%)
OS nanocomposite	O_L_ (Sn-O)	530.8	8.85	17.26
	O_v_ (vacancy)	531.3	14.58	28.44
	O_c_ (chemisorbed)	531.8	14.87	29.01
	C=O	532.6	12.96	25.27
CO nanocomposite	O_L_ (Sn-O)	530.7	10.65	17.99
	O_v_ (vacancy)	531.2	17.41	29.41
	O_c_ (chemisorbed)	531.7	18.06	30.51
	C=O	532.4	13.07	22.07
RS nanocomposite	O_L_ (Sn-O)	530.7	10.52	18.38
	O_v_ (vacancy)	531.2	16.85	29.45
	O_c_ (chemisorbed)	531.7	17.0	29.70
	C=O	532.4	12.85	22.46

**Table 4 molecules-27-08707-t004:** Summary of CH_3_COOH sensors is based on different sensing materials reported in the literature and this work.

Material	Operating Temp. (^o^C)	Concentration (ppm)	Response (%)	Response Time, *t_res_* (s)	Recovery Time, *t_rec_* (s)	Ref.
Hierarchical SnO_2_ nanoflowers	260	100	47.7	18	11	[46]
Porous flower-like SnO_2_	340	20	5.0	11	6	[47]
Mg-doped ZnO/rGO composites	250	100	200	60	35	[48]
Pr-doped ZnO nanofibers	380	400	7.38	51	40	[49]
CdS*_x_*Se_1−*x*_ nanoribbons	200	100	5.7	80	50	[50]
Mesoporous CuO	200	10	5.6	79	53	[51]
MgGa_2_O_4_/graphene composites	RT	100	363	50	35	[39]
4HQ-rGO/Cu composite	RT	500	1.75	5	5	[40]
Sn_3_O_4_-RGO-RS nanocomposite	RT	2	74	15	36	This work
Sn_3_O_4_-RGO-RS nanocomposite	RT	0.3	4	25	11	This work

RT = Room temperature.

## Data Availability

Not applicable.

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
