# Peer review of "Highly Sensitive Sub-ppm CH3COOH Detection by Improved Assembly of Sn3O4-RGO Nanocomposite"

_molecules, 2022, doi:10.3390/molecules27248707_

Round 1
Reviewer 1 Report
In this manuscript, the authors prepared Sn3O4-rGO nanocomposites for CH3COOH sensors. They used different preparation strategies to have three types of materials. The materials have been characterized. CH3COOH sensors have been commercialized for many years, I doubted whether there is a necessity for such types of sensors using regular materials like Sn3O4 and rGO without a unique discussion or specific application. I strongly suggested the authors refine the paper to reflect the novelty. There are several concerns needed to be solved before the paper can be accepted for publication in Molecules.
1. Sensitivity and selectivity are both important for gas sensors, the authors should evaluate the selectivity of their sensors.
2. Actually, the hydrothermal parameters are similar. The only difference was the order of the addition of chemicals or materials. However, there is no discussion on how the addition order changed the nanocomposites and the sensing performance. I think this would be the key point that makes this work distinguishable from others.
3. Why the profile of the curve of RS in Figure 8c is different from the curve in Figure 9b?
4. What is the mass raio of Sn3O4 to rGO in each sample, does it affact the sensing performance?
Author Response
Dear Dr. Zhong and Honoroble Reviewers,
Thank you for your positive feedback on our manuscript ‘Highly Sensitive Sub-ppm CH3COOH Detection by Improved Assembly of Sn3O4-RGO Nanocomposite’. We would like to submit a revision after accounting for reviewers' feedback. The changes in the manuscript can be found in red text, while we are replying to each comment below in blue text. Please see the attachment.
Thank you for your commitment. We wait for your decision.
Regards,
The authors

Reviewer 2 Report
Comments:
- I appreciate that the reference literature is up to date, and there is no self-citation.
- The OS composite is prepared by the in-situ decoration of graphene oxide sheets with Sn3O4 nanoparticles. It results in more uniformly decorated composite sheets, as seen in TEM image of it. This uniform decoration may be possible because of the characteristic features of graphene oxide sheets, including their functionality and well-separated individual graphene layers. However, uniform decoration is impossible in the RS composite because of the loss of characteristic functionality in RGO. TEM images of RS justify it too. However, despite the uniform decoration in OS, RS shows higher sensing performance. How will the authors justify the observation? Is that because of the double reduction of RGO? If so, do the authors think the prolonged reduction of OS composite will improve its performance?
- The crystallite size value of CO is less than the other two composites. However, the particle size obtained from TEM images does not show much deviation from the other two composites. How do the authors justify the observation?
- In the XPS data (Table1), the RS composite has less carbon percentage than OS, even though RGO undergoes a prolonged reduction in the RS composite. Justify?
Suggestion:
In line 390 the authors given that’…. the spherical 3D structure of nanoparticles…’. Therefore, it will be good if the authors can support the statement with a FESEM image of the composite.
Author Response

(The authors gave the same response as above.)

Round 2
Reviewer 1 Report
The authors have addressed all my concerns. The manuscript can be accepted for publication as it is.